# Spirituality and Employment in Recovery from Severe and Persistent Mental Illness and Psychological Well-Being

**DOI:** 10.3390/healthcare9010057

**Published:** 2021-01-07

**Authors:** Jesús Saiz, María Galilea, Antonio J. Molina, María Salazar, Tiffany J. Barsotti, Deepak Chopra, Paul J. Mills

**Affiliations:** 1Department of Social, Work and Differential Psychology, Complutense University of Madrid, 28223 Madrid, Spain; magalile@ucm.es (M.G.); antmolin@ucm.es (A.J.M.); msalaz05@ucm.es (M.S.); 2Department of Family Medicine and Public Health, University of California San Diego, La Jolla, CA 92093, USA; tiffany@healandthrive.com (T.J.B.); nonlocal@deepakchopra.com (D.C.); pmills@health.ucsd.edu (P.J.M.)

**Keywords:** spirituality, recovery, severe and persistent mental disorder, employment, psychological well-being

## Abstract

People diagnosed with severe and persistent mental illness (SPMI) face multiple vulnerabilities, including when seeking employment. Among SPMI patients, studies show that a stronger sense of spirituality can help to reduce psychotic symptoms, increase social integration, reduce the risk of suicide attempts and promote adherence to psychiatric treatment. This study examined how the variables spirituality and employment affect the recovery process and psychological well-being of people with SPMI who attend employment recovery services. The sample consisted of 64 women and men diagnosed with an SPMI. The assessment instruments included the Recovery Assessment Scale, Ryff Psychological Well-Being Scale, Work Motivation Questionnaire, Daily Spiritual Experience Scale, and Functional Assessment of Chronic Illness Therapy—Spiritual Well-Being (FACIT-Sp12). Hierarchical regression analyses were performed to compare three different models for each dependent variable (recovery and psychological well-being). The findings showed that job skills predicted psychological well-being and recovery. When spiritual variables were included in the model, job skills dropped out and the dimension meaning/peace of the FACIT-Sp12 emerged as the only significant predictor variable. Integrating spirituality into recovery programs for people with SPMI may be a helpful complement to facilitate the recovery process and improve psychological well-being.

## 1. Introduction

In the United States, 20% of adults have or have had a common mental health problem (51.5 million in 2019), and 5.2% have a serious mental illness (13.1 million in 2019) [1]. These numbers represent an increase of mental health problems from just 11 years ago. In Spain, for example, in 2009, 9.6% of the population over 15 years old suffered from a mental health problem, while in 2017, this number had risen to 10.8% [2].

Severe and persistent mental illness (SPMI) is a complex concept that fundamentally covers three areas [3]: diagnosis, disability and duration. Goldman et al. [4] (p.22) defined SPMI patients as “persons who suffer certain serious psychiatric and chronic disorders including schizophrenia, bipolar affective disorder, organic brain syndromes, paranoid disorders and other psychoses, as well as serious disorders of the personality which impede or prevent the development of their functional capacities in relation to daily life aspects, such as personal hygiene, self-care, self-control, interpersonal relations, social interactions, leisure activities, and work. These later conditions also impede the development of their economic self-sufficiency. In addition, many of these persons have been hospitalized at some time in their lives, changing the duration of their condition”. 

In this way, people with SPMI are in a vulnerable situation in which they find serious difficulties when performing certain activities, such as obtaining a job. Having a job is beneficial for mental health, offering personal empowerment and the construction of a life project. The process of obtaining a job for a person suffering from SPMI can be very difficult. In order to facilitate this process, there are specific recovery services for people with SPMI [5], including employment services [6].

### 1.1. Recovery from Persistent Mental Disorders and Psychological Well-Being

According to Anthony [7], recovery from a persistent mental disorder is “a unique, personal process, of changing attitudes, values, feelings, goals, abilities, and roles. It is a way to live a satisfying life, with hope and contribution, even with the limitations caused by illness. Recovery implies developing a new meaning and purpose in life, as the person grows beyond the catastrophic effects of mental illness…” (p. 527).

The core values in a recovery process have been identified as [8]: the involvement of the person; the need to increase self-esteem; the potential for growth, promoting and generating hope; orientation toward the person, focusing on strengths and abilities, not on difficulties; and self-determination and self-choice as a cornerstone during the recovery process. Other authors have also added [9]: meaning of life, identity, empowerment and hope and optimism about the future. 

It is important to understand that recovery does not mean a clinical cure, but rather it is a change in order to build a life beyond the disease. According to Andresen, Oades and Caputi [10], there are a series of essential elements to complete this process: Establishment of a positive identity, with a positive sense of self that incorporates the disease;Building a meaningful life;Finding hope and keeping it, pursuing the goal of believing in yourself, achieving self-control and having an optimistic vision of the future;Assuming responsibility and control with the disease and with life.

There are many approaches to studying well-being. Keyes et al. [11] defined subjective well-being (p. 1007) as “the evaluation of life in terms of satisfaction and balance between positive and negative affect”, and psychological well-being (p. 1007) as “the perception of engagement with existential challenges of life”. Spiritual well-being has been described as a “dynamic and affective dimension of religion and spirituality that impacts the way that people experience, understand and live their lives” [12] (p. 2). In relation to psychological well-being and recovery, in a recent study carried out on people diagnosed of SPMI in recovery [13], it was found that those who were in more advanced stages of recovery scored higher in psychological well-being. 

In this current study, we focus on psychological well-being and spiritual well-being, as they have been shown to maintain a close relationship with recovery.

### 1.2. Employment and Recovery from Persistent Mental Disorders

It is well known that work for people diagnosed with SPMI is beneficial in several aspects [14]. For instance, it (1) provides a feeling of normality, acceptance, belonging and fulfilment of norms and values; (2) gives structure, energy and a balanced daily life; and (3) increases well-being and strengthens one’s identity.

Lagerveld et al. [15] found that work-focused treatment improved outcome measures on duration until return to work, mental health problems, and costs to the employer in 89 clients with common mental disorders (depression, anxiety or adjustment disorder) in comparison with 79 clients that received only regular cognitive–behavioral therapy. Other authors [16] have reported that for people in recovery from SPMI, work had personal meaning and promoted recovery. Specifically, they found that work fostered pride and self-esteem, offered financial benefits, provided coping strategies for psychiatric symptoms and ultimately facilitated the process of recovery. Naranjo-Valentín et al. [17], showed that people with SPMI who had attended public services for employment and received training in skills such as job searching, CV preparation and interview skills had more success finding employment and progressed in their recovery process.

Finally, the inclusion of employment services in recovery-oriented programs has been suggested not only for people with SPMI, but also in addictive behavior recovery [18]. 

### 1.3. Spirituality and Recovery from Persistent Mental Disorders

Despite the large number of definitions offered for religiosity and spirituality, there is a relative consensus around understanding the terms as defined by Cook [19], who suggests that spirituality goes beyond religion, with a focus on interpersonal relationships and what they entail and including consciousness, meaning and purpose to life, self-knowledge, humanity, transcendence, values, authenticity, love and compassion. Spirituality appears in individuals’ inner existence and in social groups. Cook divides spirituality into three dimensions: intrapersonal (the connection of the subject with themselves), interpersonal (provides others with solidarity, understanding or acceptance) and transpersonal (search of the transcendence) [20]. In this study, we adhere to Cook as well as other authors [21] to understand spirituality as a concept that includes religion, being broader than this and encompassing questions of the meaning of life and relationships between the transcendent and the sacred.

The mechanisms in which spirituality affects health have been explored by various models and theories [22,23], and although much work remains to be done to fully explain this relationship, Koenig’s model [24] suggests spirituality as a resource that promotes health and well-being through positive psychological states, meaning of life, connection with others, harmony, hope, tranquility and moral values [25,26]. For instance, in patients with chronic diseases, it has been observed that more spirituality is associated with better clinically related symptoms such as depressed mood and anxiety, emotional variables (affect, anger), well-being (optimism, satisfaction with life), and physical-health-related outcomes (fatigue, sleep quality) [27]. Other authors [28] have found that lack of spirituality leads to lower levels of mental health.

Regarding the role of spirituality in the recovery process, some authors [29,30,31] have underlined the importance of taking spirituality into account in the process of recovery from SPMI. There is evidence that among SPMI patients, spirituality in most cases can help in recovery, reducing psychotic symptoms, increasing social integration, reducing the risk of suicide attempts and promoting adherence of psychiatric treatment [32]. Furthermore, within the model of the five recovery stages proposed by Andresen et al. [10], authors [13] have identified the specific importance of gratitude in the rebuilding stage of recovery (4th stage); compassion, for awareness (2nd stage) and growth (5th stage); inner peace in the growth stage (5th stage); and connection with life in the stages of awareness, preparation, rebuilding and growth of the recovery process (2nd to 5th stages).

However, despite these advances, it is still necessary to contextualize spirituality within the process of recovery from SPMI and evaluate its influence along with other variables of proven recovery value, such as employment. For this reason, this study aims to understand how the variables spirituality and employment affect the recovery process and the psychological well-being of people with SPMI who attend to employment recovery services.

The following hypotheses are tested:

**Hypothesis 1** **(H1).**
*Recovery will maintain a highly significant positive correlation with psychological well-being.*


**Hypothesis 2** **(H2).**
*Employed people with SPMI will score higher in recovery and psychological well-being.*


**Hypothesis 3** **(H3).**
*Spirituality will correlate positively and significantly with the recovery process and psychological well-being of people with SPMI.*


**Hypothesis 4** **(H4).**
*Spirituality and employment will be predictors of the recovery process from SPMI and psychological well-being.*


## 2. Materials and Methods 

### 2.1. Study Design and Procedure

To test our hypotheses, we performed an exploratory, ex post facto, simple retrospective study [33]. This project involved strategic collaboration with two employment recovery services within the public network for people with SPMI of the state of Madrid, Spain (CRL Nueva Vida and CRL Coslada). We used participants only from these specific employment services in order to explore the role of employment in the recovery of people with SPMI more deeply.

We used convenience sampling, as participants were contacted by professionals at the aforementioned centers, who were requested by email for an appointment date on which they could complete our questionnaires. Questionnaires were administered by a psychologist with experience in recovery for people with SPMI, in the same centers participants already attended. Before administration of the questionnaires, the study was briefly explained, and participants were assured of their anonymity and the confidentiality of their information before providing written informed consent.

Participants took between 20 and 30 min to complete the questionnaires. The questionnaires were administered in small groups so, if needed, the participants could ask the psychologist any questions they may have. They were informed that they could stop filling out the questionnaire if they felt uncomfortable. Only one of the study participants left the questionnaire incomplete because of this issue. The study period was from January 2020 through May 2020. The study received the approval of the Deontological Commission of the Faculty of Psychology of the Complutense University of Madrid (2020/21-016).

### 2.2. Sample

The sample consisted of 64 people diagnosed with a SPMI, who attended two employment recovery services. Employment recovery services are for a high profile of consumers, and they must have several cognitive and social skills and job interest. These two services were attended by people with the following diagnoses: schizophrenia (46.75%), other psychotic disorders (27.13%), personality disorders (16.91%) and mood disorders (9.21%). They were 39 men and 25 women. The age range was between 24 and 58 years old (M = 43.38, SD = 9.12) (see Table 1).

The inclusion criteria were: Diagnosis of SPMI;Age greater than 18 years;Attending an employment service for people with SPMI;Accepting voluntary participation through informed consent; andPsychopathological stability confirmed by service staff.

### 2.3. Measures

#### 2.3.1. Recovery and Psychological Well-being

There are several scales to evaluate recovery, some of which provide information about the stages of recovery, such as the Stages of Recovery Instrument [34], and others offer information about the recovery and its related factors [35]. Here, since we were interested in getting a comprehensive measure of recovery, we chose the Recovery Assessment Scale (RAS) [36]. The RAS is a widely validated scale which includes the assessment of five different recovery factors for people with SPMI, with a special focus on hope and self-determination. In addition, this test has been previously validated and adapted to Spanish [37,38]. The RAS includes 41 items, uses a 5-point Likert scale that ranges from 1 = “totally disagree” to 5 = “totally agree” and groups these items into five factors: personal confidence and hope (e.g., “I have a purpose in life”); willingness to ask for help (e.g., “I ask for help when I need it”); goal and success orientation (e.g., “I have a desire to succeed”); reliance on others (e.g., “I have people I can count on”); and not being dominated by symptoms (e.g., “My symptoms interfere less and less in my life”). The RAS offers a final score by obtaining the average of the scores of the five factors, where the higher the score the better recovery. The Cronbach’s Alpha obtained for the current sample was 0.90.

To explore psychological well-being, the adaptation to Spanish [39] of the Ryff psychological well-being scale (PBW) [40,41] was used. It comprises 39 items distributed in 6 dimensions (self-acceptance, positive relationships, autonomy, environmental mastery, personal growth and purpose in life) and on a 6-point Likert-type scale (from “Totally disagree” to “Strongly agree”). The subscales have a reliability greater than 0.70. In order to reduce the time to complete all of the questionnaires, only the dimensions corresponding to autonomy (e.g., “I have confidence in my opinions even if they are contrary to the general consensus”), personal growth (e.g., “I have the sense that I have developed a lot as a person over time”) and self-acceptance (e.g., “the past had its ups and downs, but in general I wouldn´t change it”) were applied in this research. These three dimensions were selected to assess well-being in relation to the ability to maintain individuality and independence in different social settings; positive attitude toward the self, associated with self-esteem and self-knowledge; as well as the idea of learning and positive evolution of the person. The final score was obtained as the average of the scores of the three factors, where a higher score indicated better psychological well-being. The reliability of the measure for the sample used here was 0.82.

#### 2.3.2. Employment

First, participants were asked directly if they had a job or not, coding Yes = 1 and No = 2.

As a second measure of employment, and with the aim of knowing the employment motivations of the participants, the Work Motivation Questionnaire (WMQ) for people with SPMI was also applied [42]. This is an instrument that evaluates the degree of motivation for work and integration into the world of work and differentiates between eight motivational factors: job satisfaction, integration into the work environment, social acceptance, social performance, job skills, self-esteem, perception of family support and job assertiveness. It consists of a total of 38 items, in a dichotomous response scale (“true” or “false”), where true was coded as 1 and false as 2. The total scores were obtained by adding the items of each factor. Only when necessary, the responses were recoded, and higher scores meant more work motivation. For this study, in order to reduce the response time of the questionnaires and specifically explore the motivation for employment, the following dimensions were used (26 items): job satisfaction (e.g., “When I work I feel better”), integration into the work environment (e.g., “Changes in working conditions make me nervous”), job skills (e.g., “You are likely to have problems with punctuality”) and job assertiveness (e.g., “I would never do similar jobs where I had problems again”). Regarding the people that were unemployed at the application time, we asked them to respond in a hypothetical way, as if they were employed, recalling previous employment when it was possible. The Cronbach’s alpha for the original test is 0.87 and for the version used here was 0.71.

#### 2.3.3. Spirituality

There are numerous scales to assess spirituality in health contexts [43]; in order to provide an assessment as broad as possible, without making it too time-consuming for the participants, we used two complementary instruments. One of them refers to behavioral aspects and the other explores spiritual well-being from cognitive and affective approaches.

The Daily Spiritual Experience Scale (DSES) [44] was designed to quantify ordinary experiences of spirituality. It is made up of 16 items where the first 15 explore the frequency of different spiritual experiences, using a 6-option Likert-type scale (ranging from “many times during the day” to “never or almost never”), while item 16 offers 4 possible answers (“not at all”, “something close”, “very close” and “as close as possible”). The DSES encompasses elements such as connection, transcendent sense of self, strength and comfort, peace, divine help, perception of divine love, admiration, gratitude and appreciation, compassionate love, union and closeness to the divine (e.g., “During worship, or at other times when connecting with God, I feel joy which lifts me out of my daily concerns”; “I feel a selfless caring for others”). The final score was obtained by adding the scores from all the items. Higher scores indicate a higher level of spirituality in one’s life. The validation into Spanish [45] obtained a Cronbach’s alpha of 0.91, and for the current sample, it was 0.96.

The Functional Assessment of Chronic Illness Therapy—Spiritual Well-Being (FACIT-Sp12) [12] was chosen to evaluate spiritual well-being. This test includes 12 Likert-type items with 5 response options (ranging from “not at all” to “a lot”). The FACIT-Sp12 assesses three dimensions of spirituality: meaning, peace and faith, which group into two factors: meaning/peace and faith, the first including 8 items and the second 4. Meaning is based on a cognitive component (e.g., “My life has been productive”), while peace (e.g., “I feel a sense of harmony within myself”) and faith (e.g., “I find strength in my faith or spiritual beliefs”) are considered more affective components. The Cronbach’s alpha of the FACIT-Sp 12 in the Spanish adaptation [46,47] is 0.85, and for the current sample, it was 0.79.

Additionally, gender, age and education degree were taken into account. Gender was coded as female = 2 and male = 1; age was coded in actual years; and education degree included the following categories: “without studies”, “elementary studies”, “secondary studies”, “high school or vocational training” and “university studies”.

### 2.4. Data Analysis

First, reliability analyses were performed, obtaining the Cronbach’s alpha coefficient to assess the internal consistency of the instruments used in this study. Subsequently, the assumptions of normality of the data were verified by means of the Levene test; since these were fulfilled, Student *t*-tests and analysis of the variance (ANOVA) of one factor were performed.

Next, in order to investigate the relationships between recovery and psychological well-being and motivation for employment and spirituality, Pearson’s bivariate correlation analyses were performed.

Finally, hierarchical regression analyses were performed to compare three different models for each dependent variable (recovery and psychological well-being). Model 0 included age and gender; Block 1 included age, gender, employment, job satisfaction, integration into the work environment, job skills and job assertiveness; and Model 2 included all the previous variables plus DSES, FACIT-Sp12-Meaning/Peace and FACIT-Sp12-Faith. Those whose *p*-value was less than or equal to 0.01 were taken as definitive models. The tolerance level of the independent variables is reported as a control measure for the degree of collinearity, assuming values less than 0.10 as problematic. SPSS 25 (SPSS Inc., Chicago, IL, USA) was used for the statistical analysis.

## 3. Results

As shown in Table 1, it was observed that the highest percentage of participants (42.2%) had completed university studies, 32.8% vocational training or high school, 14.1% secondary studies and 10.9% elementary studies. Likewise, it was found that 54.7% had a job. There were no differences in recovery and psychological well-being by gender, nor did we find significant correlations between age and the recovery and psychological well-being. Although the ANOVA indicated that there were significant differences in recovery according to the studies of the person (*F* [3, 60] = 2.81, *p* < 0.05), post hoc tests did not reveal any clear difference between the study groups, nor were differences found for the psychological well-being variable.

Finally, as can be seen in Table 2, consistent with what was proposed in the first hypothesis, the measures of psychological well-being and recovery showed a highly significant correlation (r = 0.70, *p* < 0.001).

### 3.1. Employment and Recovery

Contrary to what was established in the second hypothesis, we did not find significant differences in recovery between people who had a job and those who did not, *t* (59) = 0.27, *p* = 0.786. Similarly, we did not find differences in their psychological well-being, *t* (62) = 0.41, *p* = 0.681. In addition, we did not find differences in the WMQ dimensions between people who had a job and those who did not (work environment, *t* (62) = −0.92, *p* = 0.359; job skills, *t* (62) = −0.38, *p* = 0.700; job assertiveness, *t* (62) = 0.79, *p* = 0.431; job satisfaction *t* (62) = 0.84, *p* = 0.403).

Alternatively, as shown in Table 2, we found significant correlations in the variables of integration to the work environment, job skills and job assertiveness with the variables of psychological well-being and recovery, with the variable job skills showing the higher correlation with recovery (r = 0.477, *p* < 0.001) as well as with psychological well-being (r = 0.392, *p* < 0.01). Job satisfaction was the only employment dimension that did not correlate with measures of recovery or psychological well-being.

### 3.2. Spirituality and Recovery

Consistent with what was established in the third hypothesis, and as can be observed in Table 2, all the spirituality variables showed significant correlations with the recovery and psychological well-being measures, highlighting the strongest correlation of the meaning/peace dimension both for psychological well-being (r = 0.722, *p* < 0.001) and for recovery (r = 0.841, *p* < 0.001).

### 3.3. Predictive Models of Recovery and Psychological Well-Being

As is shown in Table 3, the regression analysis procedure showed that, considering recovery a dependent variable, only when employment variables were added was the variable most relevant to explain recovery job skills (22.8% of the variance explained; *p* < 0.001). Alternatively, when variables related to spirituality were introduced, the only variable that emerged in the model was the meaning/peace dimension of spiritual well-being, explaining 70.8% of the variance; *p* < 0.001.

We found very similar results when the predictive variables of psychological well-being were explored (see Table 4). For the model that included only the employment variables (Model 1), the most relevant variable was job skills (15.4% of the variance explained; *p* < 0.001). Additionally, when variables related to spirituality were introduced (Model 2), the variable that emerged was the dimension meaning/peace of spiritual well-being, explaining 52.1% of the variance; *p* < 0.001.

Thus, the findings partially confirmed the fourth hypothesis since although the employment variables explained a significant percentage of recovery and psychological well-being, when all the variables were included, the dimension of meaning/peace of spiritual well-being was the only predictor for the recovery and psychological well-being models. Finally, this variable presented a better fit for the recovery model than for the psychological well-being one.

## 4. Discussion

This study aimed to explore how the variables of spirituality and employment affect the process of recovery and psychological well-being in people with SPMI who attend an employment recovery service. The data showed that job skills are important elements to consider in recovery and psychological well-being, but that the search for peace and meaning is even more important for people who are in a recovery process.

This study presents some limitations that must be recognized. The sample, possibly due to difficulty of accessing potential participants, was limited in number. Additionally, due to the nature of our study, we selected participants only from employment services for people in recovery from SPMI, who presented a very specific profile (e.g., high levels of educational attainment, cognitive and social skills, job interest). In future studies, it would be appropriate to broaden the scope of recovery services (housing services, daily center services, and so on), include other important variables of recovery different than employment (such as social support or community participation) and increase the number of participants. Further, quantitative research for a subject as sensitive as spirituality in people suffering from SPMI obliges researchers to establish precautions, such as reducing items to avoid fatigue, administering the study in small groups to provide sufficient explanations or discontinuing the tests if someone feels uncomfortable with them. Therefore, we had to limit the scope of our instruments, such as reducing the dimensions of the PWB and the WMQ. In line with this, we did not collect more data about the participants’ employment history, and this should be explored in more depth because it could have some impact on the results too. Finally, our participants were Spanish individuals who used public services for people with SPMI, so the generalization of our findings is culturally limited. Nevertheless, considering these limitations, we might extract from our data significant information for the understanding of spirituality in recovery process.

In the first place, our findings partially coincide with another study [48] that, when evaluating the predictive variables of a recovery process in people with a diagnosis of schizophrenia, found adaptive coping strategies among the most significant variables, such as problem solving, as well as participation in socially valued activities. In this way, helping people in recovery to acquire job skills that allow them to participate more effectively and adaptively in productive environments can be an important strategy to improve their recovery and psychological well-being.

Additionally, this study extends what has been found in previous research [13] when describing the effect of spirituality in different stages of recovery. Thus, even in people who are in employment services for the recovery of people with SPMI, it seems that spirituality could have greater importance in their own recovery and psychological well-being than other classic elements of recovery, such as a job itself. The most significant aspects of spirituality were those that implied spiritual well-being, especially in terms of peace and meaning with life. Hypothetically, this could be because recovery, as defined by Anthony [7], implies developing a new meaning and purpose in life and living a satisfying life, kindly incorporating the catastrophic effects of mental illness. These results go in the same direction as other authors´ [49], who previously signaled that failure to address meaninglessness can lead to psychopathologies such as depression, anxiety, addiction, lower levels of well-being and suicide. Accordingly, applying meaning-centered therapies [50] might also be a good strategy to facilitate the recovery of people with SPMI. However, coinciding with Saiz et al. [13], here, it was also found that behavioral components of spirituality showed a positive correlation with psychological well-being and recovery, suggesting that daily spiritual experiences are also helpful for people in recovery.

Finally, of the four job motivations we explored (job satisfaction, integration into the work environment, job skills and job assertiveness), we found that job satisfaction did not show a correlation with psychological well-being nor recovery. This could suggest that, as pointed out by other authors [16], if employment is mediated by other elements such as pride and self-esteem, it may represent an important element for recovery, regardless of the degree of satisfaction of people with it.

Considering all of this, for the future, it is recommended to design interventions aimed at improving SPMI patients’ spirituality, which could be experimentally evaluated in randomized controlled studies and empirically verified by examining the efficacy of spirituality in the recovery of SPMI. Lukoff [51] already noted that in the United States, mental health systems are undergoing a “quiet revolution”, as mental health providers, government agencies, consumers, and other advocates work together to introduce spirituality into the field of mental health. Maybe it is time to take a step further and consolidate a holistic mental health system which considers human beings in the most complete way.

## 5. Conclusions

For the recovery of people diagnosed with SPMI, attaining adequate job skills is an important element. However, providing these patients with peace and meaning in their lives turns out to be a more significant element, both for their recovery and for their psychological well-being. Integrating spirituality into recovery programs for people with SPMI seems to be a necessary complement for facilitating the recovery process and improving their psychological well-being.

## Figures and Tables

**Table 1 healthcare-09-00057-t001:** Sociodemographic variables.

Variables		*N*	%
Age ^1^		43.3	9.1
Gender	Man	39	60.9
	Woman	25	39.1
Studies	Elementary studies	7	10.9
	Secondary studies	9	14.1
	Vocational training or high school	21	32.8
	University studies	27	42.2
Employment	Yes	35	54.7
	No	29	45.3
Total		64	100.0

^1^ Mean and Standard Deviation (SD).

**Table 2 healthcare-09-00057-t002:** Correlations between the studied variables.

Variables	Psychological Well-Being	Recovery
Psychological well-being		0.700 *** [0.569, 0.789]
Recovery	0.700 *** [0.569, 0.789]	
FACIT-Sp12-meaning/peace	0.722 *** [0.597, 0.811]	0.841 *** [0.769, 0.891]
FACIT-Sp12-faith	0.451 *** [0.202, 0.609]	0.544 *** [0.357, 0.696]
DSES	0.507 *** [0.300, 0.658]	0.534 *** [0.347, 0.673]
WMQ-job integration	0.296 * [0.137, 0.484]	0.334 ** [0.126, 0.516]
WMQ-job skills	0.392 ** [0.209, 0.556]	0.477 *** [0.316, 0.640]
WMQ-job assertiveness	0.252 * [0.058, 0.504]	0.353 ** [0.153, 0.541]
*N*	64	61

Note: Only significant differences are shown. Values in square brackets indicate the 95% confidence intervals for each correlation. FACIT-Sp12: Functional Assessment of Chronic Illness Therapy—Spiritual Well-Being; DSES: Daily Spiritual Experience Scale; WMQ: Work Motivation Questionnaire. Three participants did not respond to some items of the recovery measure. * *p* < 0.05, ** *p* < 0.01, *** *p* < 0.001.

**Table 3 healthcare-09-00057-t003:** Predictive models of recovery.

Models ^a^					*p*	95% CI		
		R^2^	B	SE		LL	UP	Tolerance
Model 1	WMQ-job skills	0.228	1.650	0.396	0.000	0.859	2.442	1.000
Model 2	FACIT-Sp12-meaning/peace	0.708	0.551	0.046	0.000	0.459	0.644	1.00

Note: ^a^ Only significant models are shown; Model 0 included age and gender; Model 1 included employment, job satisfaction, integration into the work environment, job skills and job assertiveness; Model 2 included all the previous variables and Daily Spiritual Experience Scale measure, FACIT-Sp12-meaning/peace and FACIT-Sp12-faith. FACIT-Sp12: Functional Assessment of Chronic Illness Therapy—Spiritual Well-Being; WMQ: Work Motivation Questionnaire.

**Table 4 healthcare-09-00057-t004:** Predictive models of psychological well-being.

Models ^a^					*p*	95% CI		
		R^2^	B	SE		LL	UP	Tolerance
Model 1	WMQ-job skills	0.154	1.545	0.460	0.001	0.624	2.465	1.00
Model 2	FACIT-Sp12-meaning/peace	0.521	0.540	0.066	0.000	0.408	0.671	1.00

Note: ^a^ Only significant models are shown; Model 0 included age and gender; Model 1 included employment, job satisfaction, integration into the work environment, job skills and job assertiveness; Model 2 included all the previous variables and Daily Spiritual Experience Scale measure, FACIT-Sp12-meaning/peace and FACIT-Sp12-faith. FACIT-Sp12: Functional Assessment of Chronic Illness Therapy—Spiritual Well-Being; WMQ: Work Motivation Questionnaire.

## Data Availability

The data presented in this study are available on request from the corresponding author. The data is not publicly available because the database is still being improved.

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
