# Peer review of "Spirituality and Employment in Recovery from Severe and Persistent Mental Illness and Psychological Well-Being"

_healthcare, 2021, doi:10.3390/healthcare9010057_

Round 1
Reviewer 1 Report
Saiz et al studied the effect of spirituality and employment on well-being and recovery among patients with a severe and persistent mental illness. They recruited 64 SPMI filling out a set of questionnaires. The outcome is the score on the RAS, and the PBW.
Predictors are the work motivation questionnaire where the authors used an abbreviated version with 26 items. Answers were either true, coded as 1, or false, coded as 2.
For the scale Job satisfaction lower scores should mean higher satisfaction (When I work I feel better - answer true gives a score of 1)
For the scale Work environment lower scores might mean less motivation, unclear since the example is "working with others involves an effort" - if this is true it can be interpreted both as being motivated or not
For the scale Job skills a lower score should mean less motivation, i.e. You are likely to have problems with punctuality if true = 1 but if false = 2, so higher scores seem to be better for work motivation or the scale reverse coded?
For the scale job assertiveness a lower score should mean less motivation, i.e. I would never do a similar job where I had problems again - if true = 1, so lower scores should mean less work motivation.
line 213 however states: higher scores meant more work motivation. However, the examples given (subscales) suggest that either some items are reverse-coded or for some subscales it is true that higher score = better motivation whereas for other subscales this is not true. Please clarify
Minor
line 194 I have confident - it should be either I am confident, or I have confidence
The results show that many SPMI are highly educated and 55% in their sample are currently employed. This matters, because the 45% who are not employed answered the work motivation scale more hypothetical, e.g. when I work I feel better - since they are not employed, do they substitute it with doing the chores? with child care? with applying for jobs? Is the Work motivation score different between those who are employed and those who are not? And how long are your participants unemployed / how recent have they taken on the job? These are important questions. If all started recently on a work, or recently lost their work, it helps to understand your data better.
Table 2: please provide the confidence intervals, and N = 6 for recovery must be a typo
Regarding your regression analysis, I recommend doing hierarchical regression ,i.e.
Model 0: recovery ~ demographic factors
Model 1: recovery ~ demographic factors + work motivation score
Model 2: recovery ~ demographic factors + work motivation score + spirituality
and then report the change in R square. This is more appropriate as you can compare models, then running your model 2 and 3 which do not contain the same factors. And since you do not report model 4, we do not know whether the change in R2 was significant. Given your small N it might not be!
Similar for well-being.
then report in a table the beta coefficient for each of the predictors, SE, Upper and lower confidence and p value ...
(JASP, jasp-stats.org) is a freeware statistics program that does this neatly for you and the table is in APA format
Line 316: a recovery processes <- replace with process
Your discussion is concise, and admits a range of short comings, but as you found out, meaning in life is essential for well-being, so how can we "train" it? Or can we not at all? There are initiatives not only for SPMI as also elderly people's well-being benefit from meaning in life. So social networks, community tasks etc have blossomed.
Author Response
Comment 1. Predictors are the work motivation questionnaire where the authors used an abbreviated version with 26 items. Answers were either true, coded as 1, or false, coded as 2.
For the scale Job satisfaction lower scores should mean higher satisfaction (When I work I feel better - answer true gives a score of 1).
For the scale Work environment lower scores might mean less motivation, unclear since the example is "working with others involves an effort" - if this is true it can be interpreted both as being motivated or not.
For the scale Job skills a lower score should mean less motivation, i.e. You are likely to have problems with punctuality if true = 1 but if false = 2, so higher scores seem to be better for work motivation or the scale reverse coded?
For the scale job assertiveness a lower score should mean less motivation, i.e. I would never do a similar job where I had problems again - if true = 1, so lower scores should mean less work motivation.
Line 213 however states: higher scores meant more work motivation. However, the examples given (subscales) suggest that either some items are reverse-coded or for some subscales it is true that higher score = better motivation whereas for other subscales this is not true. Please clarify
Response 1: Thank you for this comment that allows us to clarify our Measures section. First, we must state that all the responses of the four dimensions used from the WMQ, only when necessary, were recoded. In this way, at the end, always higher scores meant more work motivation. Secondly, the referee might be correct in his/her interpretation of the particular item from the Work environment sub-scale, but since this is an already validated instrument we choose not to change any item. In order to avoid misunderstandings from the readers, we have changed the example item in the manuscript to this other one: “Changes in working conditions make me nervous”.
Comment 2. Minor. line 194 I have confident - it should be either I am confident, or I have confidence.
Response 2: Thank you so much for signaling this error. We have corrected as recommended (“I have confidence”).
Comment 3. The results show that many SPMI are highly educated and 55% in their sample are currently employed. This matters, because the 45% who are not employed answered the work motivation scale more hypothetical, e.g. when I work I feel better - since they are not employed, do they substitute it with doing the chores? with child care? with applying for jobs? Is the Work motivation score different between those who are employed and those who are not? And how long are your participants unemployed / how recent have they taken on the job? These are important questions.
If all started recently on a work, or recently lost their work, it helps to understand your data better.
Response 3: Thank you for this comment that allows us to improve our Measures, Results and Limitations sections. First, we have improved the description of the WMQ application procedure by adding:
“To the people that were unemployed at the application time, we asked to respond in a hypothetical way, as if they were employed, recalling previous employments when it was possible”.
Secondly, following reviewer suggestion, we have analyzed the possible difference on Work motivation between those who are employed and those who are not employed. We did not find any differences so we declare this in the text as follows:
“In addition we did not find differences in the WMQ dimensions between people who had a job and those who did not (work environment, t (62) = -.92, p = .359; job skills, t (62) = -.38, p = .700; job assertiveness, t (62) = .79, p = .431; Job satisfaction t (62) = .84, p = .403).”
Finally, the reviewer might be right, and other variables could explain some of our results. So, as we didn´t collect more data about their employment history, we set this as a limitation too.
Comment 4. Table 2: please provide the confidence intervals, and N = 6 for recovery must be a typo
Response 4: Thank you for this comment. We have introduced the CI and corrected the typo in Table 2.
Comment 5. Regarding your regression analysis, I recommend doing hierarchical regression ,i.e.
Model 0: recovery ~ demographic factors
Model 1: recovery ~ demographic factors + work motivation score
Model 2: recovery ~ demographic factors + work motivation score + spirituality
and then report the change in R square. This is more appropriate as you can compare models, then running your model 2 and 3 which do not contain the same factors. And since you do not report model 4, we do not know whether the change in R2 was significant. Given your small N it might not be!
Similar for well-being.
then report in a table the beta coefficient for each of the predictors, SE, Upper and lower confidence and p value ...
Response 5: Thank you so much for this comment that urges us to improve our Data analysis. We agree with the reviewer and believe that the procedure he/she suggests is clearer than the one we used. So, we have made all the analysis again and change the Data analysis and Tables to follow his/her suggestion. We have introduced also corrections in the Abstract and in the Results when it was necessary.
Comment 6. (JASP, jasp-stats.org) is a freeware statistics program that does this neatly for you and the table is in APA format
Response 6: Thank you for this suggestion. As we didn´t know this software before, we will download it and start checking it to use it for future works. Thank you again for bringing this issue.
Comment 7. Line 316: a recovery processes <- replace with process
Response 7: Thank you so much for signaling this error. We have corrected as recommended.
Comment 8. Your discussion is concise, and admits a range of short comings, but as you found out, meaning in life is essential for well-being, so how can we "train" it? Or can we not at all? There are initiatives not only for SPMI as also elderly people's well-being benefit from meaning in life. So social networks, community tasks etc have blossomed.
Response 8: Thank you so much for this comment that allows us to improve our Discussion. Following the reviewer advice we have added two new references and the following paragraph:
“These results go in the same direction as other authors´ [49], who previously signaled that failure to address meaninglessness can lead to psychopathologies such as depression, anxiety, addiction, lower levels of well-being, and suicide. Accordingly, applying meaning-centred therapies [50] might be also a good strategy to facilitate the recovery of people with SPMI”.
Overall this is a very positive review and encourages us to continue working with the greatest possible rigor. Thank you so much to the reviewer for his/her kind work.
Reviewer 2 Report
Just a few very minor issues in English:
line 36 an increase of???
line 49 and the construction
line 60 not on difficulties
line 128 delete 'in understanding' ?
line 159 questionnaires task (unclear)
line 221 'time consuming' rather than 'time intensive'?
Author Response
Response: Thank you very much for your kind evaluation and your language corrections. We have made all the corrections as recommended by the reviewer.
Reviewer 3 Report
This is a well considered study of a poorly understood area of research. The authors use a satisfactory definition of "spirituality", although there remain uncertainties around concepts such as "meaning" and "peace" in this context. Nevertheless, they provide a clear summary of how the concept of spirituality has developed in psychological research, with reference to the recovery from and adaption to, various psychological disorders. They then specifically address the specific topic of employment as an element of recovery and this forms the basis of their study. I found their review of the literature concise but informative and relevant. The topic of their research is clearly of considerable importance in mental health care.
I note that the subjects of this study suffered from a wide variety of "severe and persistent mental illness" but there was no report of the numbers of patients with specific disorders and no mention of the distribution of psychological measures by diagnosis. While the number of subjects may not have been sufficient in specific diagnostic groups to provide meaningful conclusions, it would be valuable to provide what data is available. There may be significant differences in spirituality between diagnostic groups (I suggest this is likely), which may be relevant to recovery. Even a trend in statistical data would be of interest in planning further research. It may also be of interest in planning management if certain diagnostic groups were more or less receptive to interventions relating to spirituality. Further I would expect there may be differences in vocational rehabilitation between diagnostic groups (as we know exist) which might be usefully modified by measures designed to enhance spirituality. Again, this study could help point the way to appropriate research.
It would appear that the levels of educational attainment by the subjects in this study were relatively high. For patients suffering from "severe and persistent mental illness", I was surprised that 42.2% had reached tertiary study and 32.8% had reached secondary study or vocational training. This again suggests that the diagnostic distribution of the patient sample is of interest. This distribution would be unusual for a group dominated by people suffering from schizophrenia, for example. A comment by the authors regarding the impact of this distribution upon vocational rehabilitation would be of interest.
The key finding that the "meaning and peace" dimensions of spirituality explained 70% of the variance in recovery from these disorders is of considerable importance. I would appreciate comment by the authors regarding this finding and possible hypotheses.
The authors appropriately address the limitations of their research and their conclusions are well founded.
Author Response
- This is a well considered study of a poorly understood area of research. The authors use a satisfactory definition of "spirituality", although there remain uncertainties around concepts such as "meaning" and "peace" in this context. Nevertheless, they provide a clear summary of how the concept of spirituality has developed in psychological research, with reference to the recovery from and adaption to, various psychological disorders. They then specifically address the specific topic of employment as an element of recovery and this forms the basis of their study. I found their review of the literature concise but informative and relevant. The topic of their research is clearly of considerable importance in mental health care.
Response: Thank you very much for your kind comments.
- I note that the subjects of this study suffered from a wide variety of "severe and persistent mental illness" but there was no report of the numbers of patients with specific disorders and no mention of the distribution of psychological measures by diagnosis. While the number of subjects may not have been sufficient in specific diagnostic groups to provide meaningful conclusions, it would be valuable to provide what data is available. There may be significant differences in spirituality between diagnostic groups (I suggest this is likely), which may be relevant to recovery. Even a trend in statistical data would be of interest in planning further research. It may also be of interest in planning management if certain diagnostic groups were more or less receptive to interventions relating to spirituality. Further I would expect there may be differences in vocational rehabilitation between diagnostic groups (as we know exist) which might be usefully modified by measures designed to enhance spirituality. Again, this study could help point the way to appropriate research.
Response: Thank you very much for this comment, which allows us to improve our sample description. We do believe that the referee is correct when suggesting that different patients may have different needs and outcomes. As the sample came from two very homogenous services, and indeed it was too small to provide meaningful conclusions, we did not collect that specific data. Rather we collected the overall diagnosis of the people that attended to these public services. Following reviewer suggestion we have added the following:
“(…) In these two services attended people with the following diagnosis: Schizophrenia (46.75%), Other psychotic disorders (27.13%), Personality disorders (16.91%) and Mood disorders (9.21%)”.
- It would appear that the levels of educational attainment by the subjects in this study were relatively high. For patients suffering from "severe and persistent mental illness", I was surprised that 42.2% had reached tertiary study and 32.8% had reached secondary study or vocational training. This again suggests that the diagnostic distribution of the patient sample is of interest. This distribution would be unusual for a group dominated by people suffering from schizophrenia, for example. A comment by the authors regarding the impact of this distribution upon vocational rehabilitation would be of interest.
Response: Again, thank you so much for his comment that allows us to improve our sample description and our limitations.
For the sample description we have added that “(…)Employment recovery services are for a high profile of consumers, and they must have several cognitive and social skills and job interest”.
And about the limitations, we have recognized that our research sample:
“(…) presented a very specific profile (e.g. high levels of educational attainment, cognitive and social skills, job interest)”.
- The key finding that the "meaning and peace" dimensions of spirituality explained 70% of the variance in recovery from these disorders is of considerable importance. I would appreciate comment by the authors regarding this finding and possible hypotheses.
Response: Thank you so much for this comment. We have improved our Discussion by adding the following paragraph:
“Hypothetically, this could be because recovery, as defined by Anthony [7], implies developing a new meaning and purpose in life, and living a satisfying life, kindly incorporating the catastrophic effects of mental illness”.
- The authors appropriately address the limitations of their research and their conclusions are well founded.
Response: Thank you very much for the feedback. Overall this is a very positive review and encourages us to continue working with the greatest possible rigor.